

# Human head orientation and eye visibility as indicators of attention for goats (*Capra hircus*)

Christian Nawroth and Alan G. McElligott

Biological and Experimental Psychology, School of Biological and Chemical Sciences, Queen Mary University of London, London, UK

## ABSTRACT

Animals domesticated for working closely with humans (e.g. dogs) have been shown to be remarkable in adjusting their behaviour to human attentional stance. However, there is little evidence for this form of information perception in species domesticated for production rather than companionship. We tested domestic ungulates (goats) for their ability to differentiate attentional states of humans. In the first experiment, we investigated the effect of body and head orientation of one human experimenter on approach behaviour by goats. Test subjects ($N = 24$) significantly changed their behaviour when the experimenter turned its back to the subjects, but did not take into account head orientation alone. In the second experiment, goats ($N = 24$) could choose to approach one of two experimenters, while only one was paying attention to them. Goats preferred to approach humans that oriented their body and head towards the subject, whereas head orientation alone had no effect on choice behaviour. In the third experiment, goats ($N = 32$) were transferred to a separate test arena and were rewarded for approaching two experimenters providing a food reward during training trials. In subsequent probe test trials, goats had to choose between the two experimenters differing in their attentional states. Like in Experiments 1 and 2, goats did not show a preference for the attentive person when the inattentive person turned her head away from the subject. In this last experiment, goats preferred to approach the attentive person compared to a person who closed their eyes or covered the whole face with a blind. However, goats showed no preference when one person covered only the eyes. Our results show that animals bred for production rather than companionship show differences in their approach and choice behaviour depending on human attentive state. However, our results contrast with previous findings regarding the use of the head orientation to attribute attention and show the importance of cross-validating results.

Corresponding authors
Christian Nawroth,
c.nawroth@qmul.ac.uk
Alan G. McElligott,
a.g.mcelligott@qmul.ac.uk

## INTRODUCTION

Body and head orientation are an important component of social interaction and the ability to recognise different attentive states of conspecific or heterospecifics yields adaptive advantages in the contexts of predation, deception or cooperation (*Emery, 2000*).

For example, gaze directed towards an individual might be considered as threat and elicit anti-predator response (*Kummer, 1967*). American crows (*Corvus brachyrhynchos)* have been shown to adjust their behaviour based on human gaze (*Clucas et al., 2013*). When an experimenter approached the subjects, crows fled sooner when the person was directly gazing at them compared to when gaze was averted. Jackdaws (*Corvus monedula)* showed similar avoidance behaviour during an approach task. When an unfamiliar human paid attention to them, their time to approach a reward was higher compared to when the unfamiliar human looked away or had its eyes closed (*von Bayern & Emery, 2009*). *Beausoleil, Stafford & Mellor (2006)* investigated whether human staring altered the behaviour of domestic sheep (*Ovis aries*) compared with no human eye contact. They found that sheep glanced at the staring human's face more often and showed higher levels of locomotor activity. However, researchers did not find differences in fear-related behaviours. Research on wild and captive primates has shown that when they have the opportunity to steal from one of two experimenters, Rhesus monkeys (*Macaca mulatta*) and several species of lemurs choose to take food from an experimenter who was not looking at the reward (*Flombaum & Santos, 2005*; *Sandel, MacLean & Hare, 2011*). Being habituated to humans and knowing that people regularly deliver food, individuals might opt to position themselves to be seen by humans or to choose a person that pays attention towards them, expecting to receive a food reward or one with less delay. When they have to make a choice, captive chimpanzees (*Pan troglodytes*) have been found to prefer to beg from an attentive rather than an inattentive person (*Bulloch, Boysen & Furlong, 2008*).

Dogs (*Canis familiaris*) are assumed to be specifically attuned to human cues due to their domestication history as companion animals (*Hare et al., 2002*). For example, they show great sensitivity to human communicative cues, such as human pointing and gazes, to find a reward in an object choice task (*Agnetta, Hare & Tomasello, 2000*; *Riedel et al., 2008*). In these tasks, subjects have to choose between two or more containers while only one is containing food (*Miklósi & Soproni, 2006*). However, domestic goats have also been shown to use a basic human pointing gesture, but not head direction or gazes, to find a reward in this task (*Kaminski et al., 2005*; *Nawroth, von Borell & Langbein, 2015*). Similar findings have been obtained for other domestic species, such as cats, *Felis catus* (*Miklósi et al., 2005*) and horses, *Equus caballus* (*Proops, Walton & McComb, 2010*; *Proops et al., 2013*). Dogs are also able to comprehend more complex gestures and can distinguish whether information is intended for them (*Kaminski, Schulz & Tomasello, 2012*; *Lakatos et al., 2012*). They are also sensitive to the attentional stance of humans. When not allowed to obtain a reward in front of them, they disobeyed faster when the human was distracted or not looking at the dog (*Call et al., 2003*). When dogs had to choose between either a person that was looking at them vs a person that had turned their back to them, dogs approached the attentive experimenter (*Udell, Dorey & Wynne, 2011*). But here again, horses (*Proops & McComb, 2010*) and pigs, *Sus scrofa* (*Nawroth, Ebersbach & von Borell, 2013*) have been shown to express similar preferences.

We investigated the ability of goats to perceive information from human attentional states. Using a food-anticipating paradigm, *Nawroth, von Borell & Langbein (2015*, *2016*)

found that goats adapted their anticipation behaviour depending on the presence or absence of an experimenter in general and his/her head and body orientation in particular. In these experiments, an experimenter remained for 30 s in an assigned posture before delivering a reward to the tested subject, and the goats' active anticipation and standing alert behaviour were analysed. The level of subjects' active anticipatory behaviour was highest when the experimenter looked in the direction of the test subject, and decreased with a decreasing level of attention paid to the subject by the experimenter. Additionally, goats 'stared' (i.e. stood alert) at the experimental setup for significantly more time when the experimenter was present but had his head and/or body directed away from the subject. However, due to previous training, it was not clear if simple conditioned responses were responsible for the goats' change in behaviour (*Proops & McComb, 2010*). Subjects may have learned that a certain posture of the experimenter (head and body oriented towards subject) would yield a reward.

To cross-validate and extend these previous findings on goats' ability to differentiate between human attentive states, we examined the approach and choice behaviour of goats towards humans who were paying different degrees of attention to them. In an important contrast to previous research on goats, these approach and choice tasks involved little or no task-related learning prior to the actual test. In Experiment 1, goats had the opportunity to approach either an attentive or inattentive person and their approach behaviour (i.e. if they preferred to move into the 'attention window' of the human) was examined. We expected goats to move around the experimenter when the person was not paying attention towards them. Being aware that both behavioural outcomes (moving around the human or not) require different amounts of energetic costs, we subjected goats to additional choice experiments. In Experiments 2 and 3, goats had to choose between an attentive vs an inattentive person. While Experiment 2 involved no training and spontaneous choice behaviour in the field, Experiment 3 was conducted in a separate test arena to control in more detail for environmental factors. Being familiar with human presence, we expected subjects to choose in a cooperative manner (i.e. choose the human that was paying attention to them) because this would decrease a delay in potential reward delivery or grooming events.

## ANIMALS, MATERIALS AND METHODS

### Ethical note

Animal care and all experimental procedures were in accordance with the ASAB/ABS guidelines for the use of animals in research (*Association for the Study of Animal Behaviour, 2016*). The study was approved by the Animal Welfare and Ethical Review Board committee of Queen Mary University of London (Ref. QMULAWERB032015). All measurements were non-invasive, and the experiment lasted no more than 10 min for each individual goat. If the goats had become distressed, the test would have been stopped.

### Subjects, housing and general procedure

The experiments were carried out at Buttercups Sanctuary for Goats (Maidstone, UK) (http://www.buttercups.org.uk). Goats were fully habituated to human presence because
of previous research (e.g. *Baciadonna, Nawroth & McElligott, 2016*; *Nawroth, Brett & McElligott, 2016*). They were aged 2–14 years and of various breeds. Routine care of the animals was provided by sanctuary employees and volunteers. The goats had ad libitum access to hay and were not food restricted before testing. Subjects were tested from 12:00 to 16:00 in June to August 2015.

## Experiment 1: approach behaviour in the field
### Experiment 1.1: body orientation

*Procedure*
The experimenter approached a focal subject that was separated from other groups by about 5–10 m. He then proceeded to get its attention by calling 'Come here'. As soon as the goat approached, a trial started. When the goat approached to about 2 m from the experimenter, the person either stayed with his front and head oriented towards the subject ('front') or turned 180° ('back'). During both conditions, the hands of the experimenter were oriented away from the subject (behind the experimenter's back in the 'front' condition; in front of experimenter in the 'back' condition). A total of 29 trials were conducted. Five goats had to be excluded because they stopped approaching the experimenter, leaving 24 successful approach trials. Twelve subjects received the 'front' condition and the other 12 subjects received the 'back' condition. Goats were never rewarded during trials.

*Data scoring and analysis*
Approach behaviour of the goats was scored live. After a goat approached the experimenter to approximately 1 m, the experimenter scored the behaviour of the goats during the following 5 s. If a goat crossed the midline of the experimenter during these 5 s with its head (threshold line; Fig. 1A), this was scored as a 'behind approach'. If the goat remained in front of the threshold line during these 5 s, this was scored as 'front approach'. Fisher's exact test was used to compare the amount of both approach behaviours in the two different conditions.

*Results*
A strong difference in the approach behaviour between the two conditions was found for the 'front approach'. Whereas all 12 subjects in the 'front' condition stayed in front of the threshold line for 5 s, only two out of 12 subjects did so in the 'back' condition (Fisher's exact test: $N = 24$; $P < 0.0001$). Whereas none of the subjects in the 'front' condition crossed the threshold line (Fig. 1A), all 10 subjects in the 'back' condition that did not stay in front of the midline moved around the human by crossing the threshold line during the duration of 5 s ('back approach').

### Experiment 1.2: head orientation
Experiment 1.1 showed that goats alter their approach behaviour depending on whether a human is orienting his front or back towards the subjects. Using a similar setup, we here investigated whether goats also alter their behaviour depending only on the head orientation of a human.

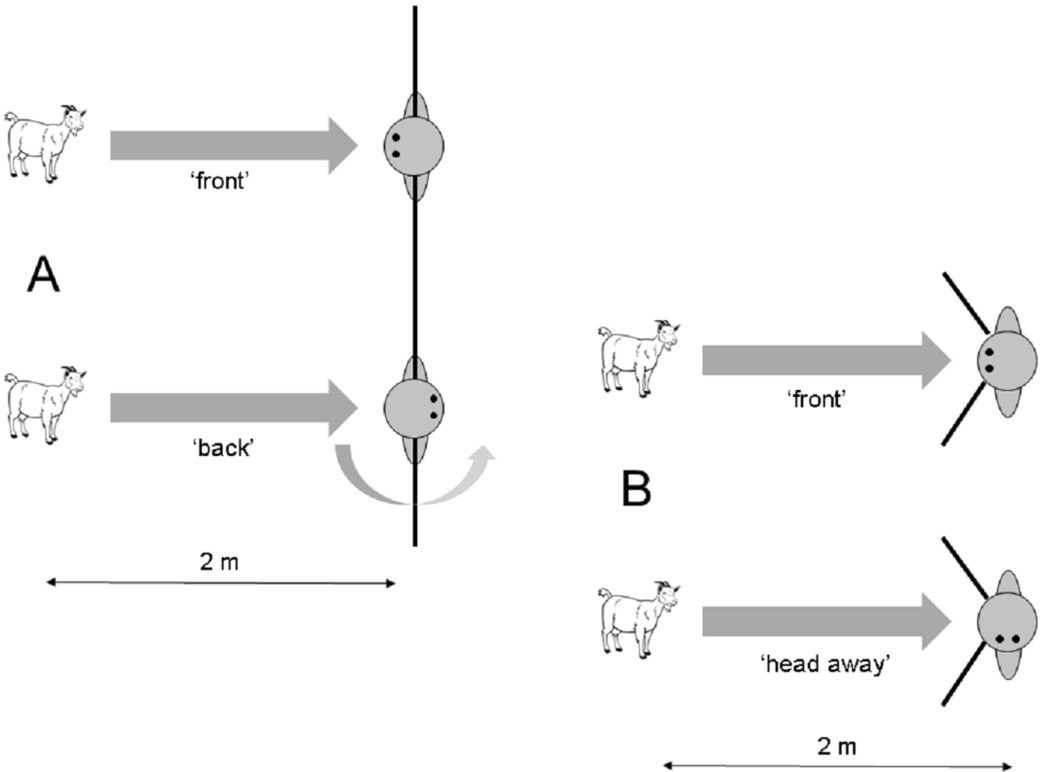

**Figure 1 Setup of the approach task in (A) Experiment 1.1 and (B) Experiment 1.2.** The black lines close to the experimenter indicate threshold lines that were used to define a specific approach behaviour. If a subject did not cross the line (and thus remained in front of it), it was scored as 'front approach'. If it crossed the line, it was scored as 'behind approach' (Experiment 1.1) or 'side approach' (Experiment 1.2).

*Procedure*

The experimenter approached a focal subject that was separated from other groups by about 5–10 m. He then proceeded to get its attention by calling 'Come here'. As soon as the goat approached, a trial started. When the goat approached as close as 2 m towards the experimenter, he either kept his front and head oriented towards the subject ('front') or turned his head 90° to the left or right ('head away'), respectively (Fig. 1B). During both conditions, the hands of the experimenter were behind his back to avoid inadvertently cueing the tested subject. A total of 27 trials were conducted. Three goats had to be excluded because they stopped approaching the experimenter, leaving 24 successful approach trials. Twelve subjects received the 'front' condition and the other 12 subjects received the 'head away' condition. Head orientation in the 'head away' condition was counterbalanced for the left and right side. Goats were never rewarded during trials.

*Data scoring and analysis*

Approach behaviour of the goats was scored live. Because goats often moved in front of the human when not rewarded instantly, we used a slightly different criterion compared to Experiment 1.1. In this experiment, the experimenter scored the behaviour of the goats in their initial approach orientation. If the goats approached the experimenter in the

middle, this was scored as 'front approach'. If the goats approached the experimenter from the left or the right side by crossing a threshold line with an angle of about 90° on each side of the experimenter, this was scored as 'side approach' (Fig. 1B). Fisher's exact test was used to compare the amount of both approach behaviours in the two different conditions.

*Results*

There was no difference in approach behaviour between the two conditions. Eleven of 12 subjects in the 'front', and 8 out of 12 in the 'head away' condition approached the experimenter from the front and not the side (Fisher's exact test: $N = 24$; $P = 0.32$). From the four subjects that approached the side of the experimenter in the 'head away' condition, only two did so for the same side the experimenter was facing. Notably, none of the goats instantly moved behind the experimenter to approach his hands in order to receive food.

## Experiment 2: choice behaviour in the field

In Experiment 1.2, goats did not differ in their approach behaviour when a human experimenter was looking at them compared to when he was looking to the side. However, goats may have just avoided additional movement or preferred to remain in front of the experimenter because they previously experienced humans moving their head frequently while living at the study site. Therefore, we conducted a second experiment, in which the energetic effort for indicating a preference for either an attentive or a non-attentive person was equal. In Experiment 2, we used a modified test paradigm to cross-validate the findings from Experiment 1. In this experiment, we used two experimenters, one paying attention to the subject while another one remained inattentive. Goats were free to approach either of the experimenters.

### Experiment 2.1: body orientation

*Procedure*

Two experimenters approached a focal subject that was separated from other groups by about 5–10 m and took position (see Fig. 2A). They then proceeded to get its attention by calling 'Come here'. As soon as the goat approached, a trial started. When the goat approached as close as 4 m to the experimenters, one experimenter turned his/her back towards the subject whereas the other experimenter remained with his/her body and head oriented to the subject. Side and identity of the attentive person was assigned in a pseudo-randomised manner. A total of 29 trials were conducted. Five goats had to be excluded because they stopped approaching the experimenter or their choice was ambiguous, e.g. standing in the middle of both experimenters, leaving 24 successful approach trials. Goats were never rewarded during trials.

*Data scoring and analysis*

Behaviour of the goats was recorded live. If a goat approached the attentive person, this choice was scored as 'correct'. Both experimenters had to agree that the focal subject approached one of them. Approach behaviour was unambiguous in all trials. To analyse

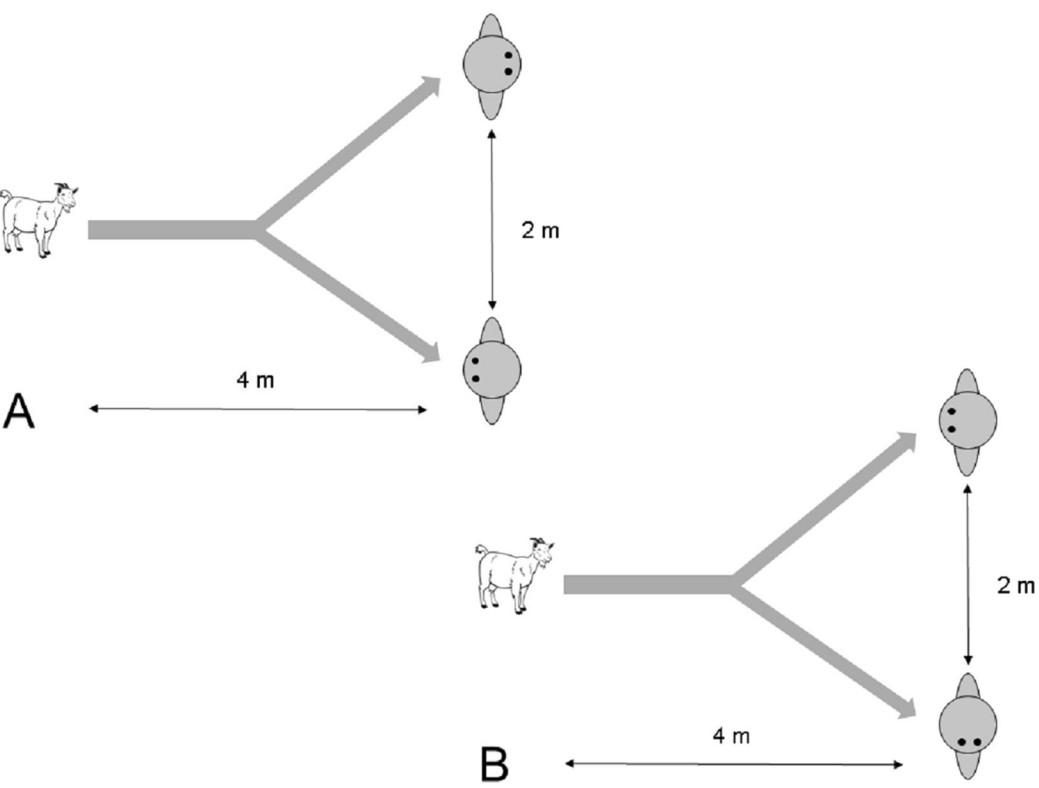

**Figure 2 Setup of the choice task in (A) Experiment 2.1 and (B) Experiment 2.2.** After a test subject started to approach the experimenters from a distance of about 4 m, one of them either turned his/her body (Experiment 2.1) or head (Experiment 2.2) away from the subject.

preferences in the choice behaviour of goats, a binomial test was conducted in which the number of approaches to the attentive person was compared to chance level (50%).

*Results*

Goats showed a preference for the attentive experimenter ($n = 24$; $K = 22$; $P < 0.001$, two-tailed). There was no side preference ($n = 24$; $K = 14$; $P > 0.5$, two-tailed) and no preference for one of the experimenters ($n = 24$; $K = 14$; $P > 0.5$, two-tailed).

### Experiment 2.2: head orientation

*Procedure*

Two experimenters approached a focal subject that was separated from other groups by about 5–10 m and took position (see Fig. 2B). They then proceeded to get its attention by calling 'Come here'. As soon as the goat approached, a trial started. When the goat approached as close as 4 m to the experimenters, one experimenter turned his/her head away from the subject whereas the other experimenter remained with his/her head oriented to the subject. All other conditions were the same as in Experiment 2.1. A total of 30 trials were conducted. Six goats had to be excluded because they stopped approaching the experimenter or their choice was ambiguous, e.g. standing in the middle of both experimenters, leaving 24 successful approach trials. Goats were never rewarded during trials.

*Data scoring and analysis*
Same as in Experiment 2.1.

*Results*
Goats showed no preference for either the attentive or the non-attentive experimenter ($n = 24$; $K = 13$; $P = 0.83$, two-tailed). There was no side preference ($n = 24$; $K = 15$; $P > 0.3$, two-tailed) and no preference for one of the experimenters ($n = 24$; $K = 13$; $P > 0.5$, two-tailed).

### Experiment 2.3: hand orientation

In Experiment 2.1, we could not rule out hand orientation as a confounding factor and thus conducted a control experiment to investigate if the position of the hands (front or back) influenced the choice behaviour of the goats.

*Procedure*
Two experimenters approached a focal subject that was separated from other groups by about 5–10 m and took position. They then proceeded to get its attention by calling 'Come here'. As soon as the goat approached, a trial started. When the goat approached as close as 4 m to the experimenters, both remained with their body and head oriented forward, while one experimenter placed his/her hands towards his/her back, whereas the hands of the other experimenter remained in front. All other conditions were the same as in Experiment 2.1. A total of 17 trials were conducted. Five goats had to be excluded because they stopped approaching the experimenter or their choice was ambiguous, e.g. standing in the middle of both experimenters, leaving 12 successful approach trials. Goats were never rewarded during trials.

*Data scoring and analysis*
Same as in Experiment 2.1.

*Results*
Goats showed no preference for either experimenter presenting the hands or the experimenter covering the hands ($n = 12$; $K = 7$; $P = 0.77$, two-tailed). There was no side preference ($n = 12$; $K = 9$; $P > 0.1$, two-tailed) and no preference for one of the experimenters ($n = 12$; $K = 7$; $P > 0.5$, two-tailed).

## Experiment 3: choice behaviour in a controlled setting

In Experiments 1 and 2, goats showed a preference for humans that oriented their body and head, but not head only, towards them. The negative results for differentiating the head orientation may be a product of confounding factors that may have influenced decision making of the goats in the previous experiments, e.g. distracting factors like environmental noise and nearby conspecifics, or their specific motivation to approach the human. Because subjects could move freely, we were not able to control for environmental disturbances and the position of conspecifics, which may have biased their decision making. In terms of their motivation, we could not exclude whether goats approached humans for other reasons than to receive a reward. For example, goats sometimes rubbed

themselves against the experimenter's clothing (C. Nawroth, 2015, personal observation), which would only require the presence but not the attention of the human. To account for this, we conducted a third and last experiment in a controlled setting. Prior to testing, subjects received food rewards while they were trained to approach two human experimenters. During test trials, subjects had the opportunity to approach one of two experimenters; one paid attention while the other was either orienting her head away from the subject or closed her eyes (*Proops & McComb, 2010*; *Nawroth, Ebersbach & von Borell, 2013*). In addition, we investigated whether goats understand the role of occlusion, by either holding a big blind in front of the face or a small blind in front of the eyes of one of the two experimenters (*Flombaum & Santos, 2005*).

*Procedure*

The experiment was carried out in a temporary test arena, which we set up within the normal daytime range of the goats. Goats were individually transferred to the test arena. During training trials, the subject was brought to the start position at the entrance of the arena, held on a leash. Two experimenters were positioned 4 m away from the starting point, kneeling, at the other side of the arena. When both experimenters took position, the goat was released from the leash and was free to approach the experimenters and to receive the reward that they offer. To prevent the goats developing a preference for one of the experimenters, the reward was administered jointly by crossing over their arms and holding out their hands together with a piece of food (pasta) in the middle of both of their hands. The experimenters also swapped sides between each trial. Eighteen subjects were given an introductory training phase in which experimenters faced forwards when giving a reward ('attentive training', Fig. 3A). Another 18 goats received a different training phase in which the experimenters were not attentive to the subjects and adopted body postures that were not repeated in the test trials ('non-attentive training', Fig. 3B). In this phase, the experimenters kneeled at 90° from the subject facing each other with their hands outstretched together and a reward held in their hands. If necessary, goats were brought on a leash to the experimenters in the first training trial. If subjects did not approach the experimenters after 1 min, a training trial was aborted. After a maximum of five training trials, all but four goats instantly and reliably approached the experimenters and thus proceeded to the test trials.

The procedure in test trials was the same as in the training trials, except that each experimenter took place in a separate corner of the arena at 5 m from the starting point and 2.5 m away from each other. Both experimenters differed in their degree of attention towards the subject. While one experimenter paid attention to the goat, the other one had her attention oriented away from the subject or her visual access was occluded. Four different test conditions were administered (Fig. 4):

a) *Head away*—both experimenters oriented their bodies to the front while one looked to the front, whereas the other one turned her head away.

b) *Eyes closed*—both experimenters oriented their bodies and heads to the front, while one closed her eyes and the other remained with eyes open.

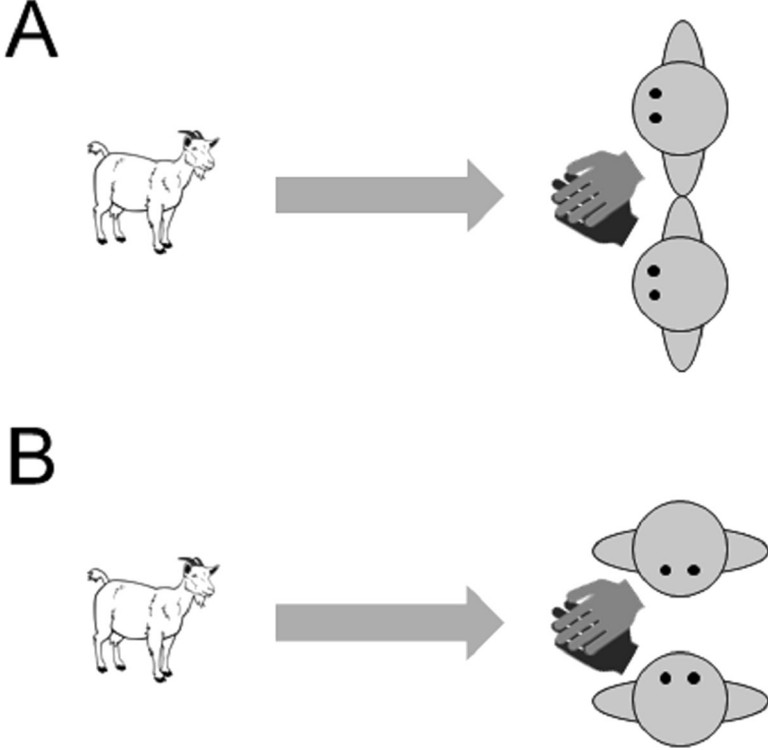

**Figure 3 Images of the two training conditions in Experiment 3 (A) attentive training and (B) non-attentive training.** A reward was administered jointly by both experimenters by crossing over their arms and holding out their hands together with a piece of food (pasta) in the middle of both of their hands.

c) *Covered head*—both experimenters oriented their bodies and heads to the front, while one covered the whole face and the other covered the chest with a blind (28 cm × 24 cm).

d) *Covered eyes*—both experimenters oriented their bodies and heads to the front, while one covered the eyes and the other covered the mouth with a blind (28 cm × 11 cm).

Each subject received one session of testing included four test trials, one for each condition. Conditions were counterbalanced for side (left/right) and experimenter (E1/E2). Goats were never rewarded during test trials. To ensure motivation, goats received a rewarded training trial after each test trial (delivered by the same experimenters as the initial training trials).

*Data scoring and analysis*

Trials were scored live and were video recorded (Sony DCR-SX33E camcorder). We analysed goat choice and latency to approach. Latency times were defined as the time between a subject's first step from the starting position and the time it approached one of the two experimenters. An approach was defined as the goat getting as close as 15 cm to one of the experimenters with the head raised towards them. A second observer coded 25% of the test trials. Inter-observer reliability for choice (Cohen's $k = 1.00$) and latency to approach (Spearman rank correlation; $r_s = 0.948$; $P < 0.001$) was excellent. Preference for the attentive or the non-attentive person, effect of experimenter identity and side

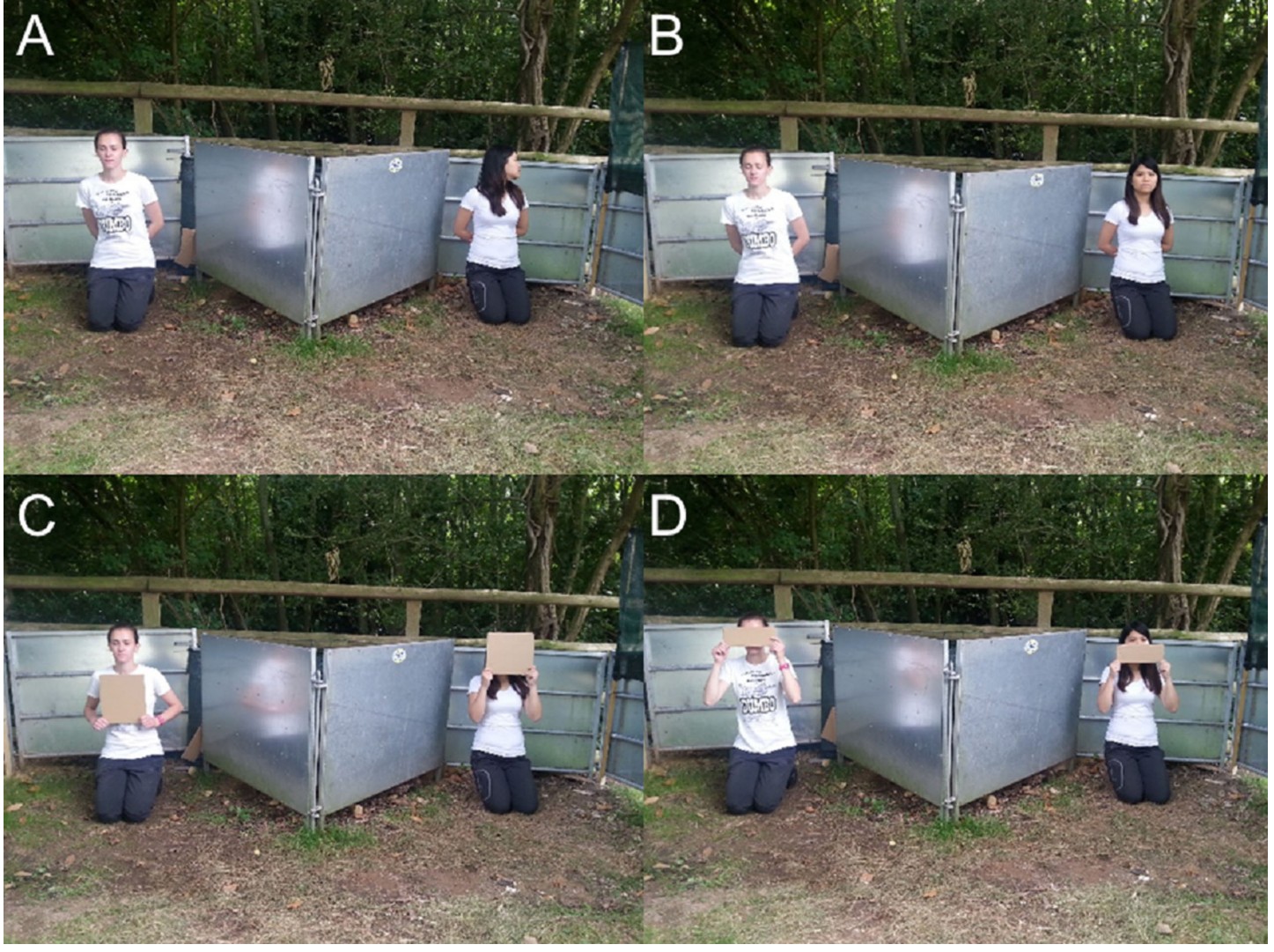

**Figure 4** **Images of the different test conditions in Experiment 3.** (A) head away, (B) eyes closed, (C) covered head and (D) covered eyes.

preferences were calculated using the binomial test for choice behaviour of goats. Effects of goat sex and of the different warm-up types on choice behaviour were analysed using Fishers exact test. Data for latency times were positively skewed and so were $\log_{10}$ transformed. The potential effects of condition and response accuracy (correct/incorrect) on the latency to approach an experimenter were analysed as fixed factors in a linear mixed model. Alpha level was set at 0.05 in all comparisons.

*Results*

Four subjects had to be excluded because they did not reliably approach the experimenters after five training trials. This left 16 subjects with the attentive and 16 subjects with the inattentive training. One of the subjects showed a strong side bias in all trials and was therefore excluded from further analysis. One subject did not make any choice after 1 min in the 'Head away' and 'Eyes closed' condition, another subject did not respond in time

**Table 1 Comparison of absolute and relative choices of goats for attentive humans given the different training trials with either attentive or non-attentive experimenters.**

| Condition | Attentive training | | Non-attentive training | | FET (*P*) |
|---|---|---|---|---|---|
| Head away | 6/14 | 43% | 10/16 | 63% | 0.46 |
| Eyes closed | 11/14 | 79% | 10/16 | 63% | 0.44 |
| Covered head | 11/15 | 73% | 12/15 | 80% | 1.00 |
| Covered eyes | 8/15 | 53% | 10/16 | 63% | 0.72 |
| Total | 36/58 | 62% | 42/63 | 67% | 0.70 |

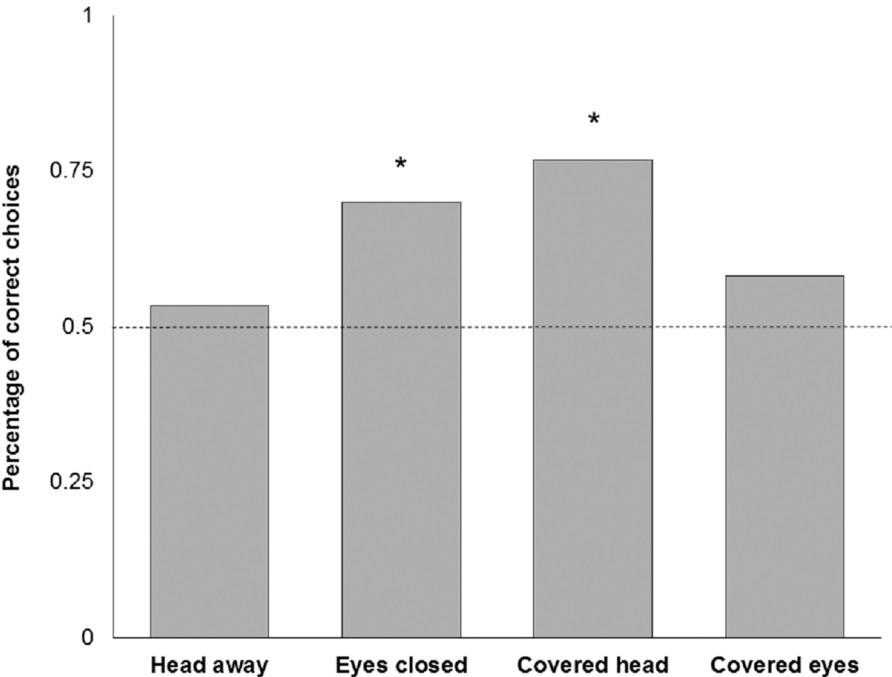

**Figure 5 Relative number of choices for the attentive person by goats in the four test conditions.**
*$P < 0.05$ (binomial test, two-tailed).

in the 'Covered head' condition. The training procedure, with either attentive or non-attentive experimenters, had no significant effect in any of the test conditions (Table 1). In test trials, goats did not show a preference for the attentive person when the other person looked away from the subject ('Head away'; $n = 30$; $K = 16$; $P = 0.856$, Fig. 5). There was a significant preference for the attentive person compared to the one that closed their eyes ('Eyes closed'; $n = 30$; $K = 21$; $P = 0.043$, Fig. 5). In addition, goats preferred to approach the attentive person compared to a person that covered the face with a blind ('Covered head'; $n = 30$; $K = 23$; $P = 0.005$, Fig. 5), but not when the other person was covering the eyes ('Covered eyes'; $n = 31$; $K = 18$; $P = 0.473$, Fig. 5). We found no effect of sex (Fisher's exact test; $P = 0.83$), experimenter identity ($n = 121$; $K = 68$; $P = 0.2$) or side ($n = 121$; $K = 63$; $P = 0.72$) on choice behaviour of goats. Latency times to approach an experimenter did not vary as a function of performance ($F_{1,110} = 1.173$; $P = 0.281$) or condition ($F_{3,106} = 0.610$; $P = 0.610$). No interaction effect was found ($F_{3,106} = 1.987$; $P = 0.120$).

## DISCUSSION

In a series of three experiments, we investigated whether domestic goats that are habituated to human presence show a preference to approach an attentive compared to a non-attentive person. Experiments 1 and 2 took advantage of the general approach behaviour of goats towards humans at the study site. The results of these experiments show that goats actively try to position themselves where a human is likely to notice them, and prefer to approach an experimenter who orients his/her body towards them. However, they showed no preference in their approach and choice behaviour when only the head orientation was given as a cue for the attentional stance. Experiment 3 was conducted in a more controlled setting and confirmed the lack of preference when only the head orientation of a human was altered. In addition, goats in this last experiment preferred to approach a person whose face was not occluded and whose eyes were not closed, indicating that they rely on cues other than head orientation when attributing attention. Contrary to previous studies in horses and pigs that reported an effect of test condition and/or performance on approach latencies (*Proops & McComb, 2010*; *Nawroth, Ebersbach & von Borell, 2013*), we did not find a difference in the approach times of goats in our experiment. Our results show that animals domesticated for production rather than companionship show differences in their approach and choice behaviour depending on human attentive states (*Hare et al., 2002*; *Nawroth, Brett & McElligott, 2016*).

Contrary to previous research, all three experiments indicate that goats do not pay attention to the head direction of a human experimenter. While it has been shown that goats decrease their anticipatory behaviour in a food-anticipation paradigm when a person is directing their head away from them (*Nawroth, von Borell & Langbein, 2015*, *2016*), we could not find behavioural differences in our subjects. It is possible that this non-preference might be due to their specific experience with humans. Goats at our study site interact with humans on a daily basis and are used to receiving food directly, although the human head is not necessarily in the direction of the goat at that time. This might have altered their general anticipation of receiving food when approaching humans that differ in their attentional stance. Alternatively, goats may not have interpreted the human with its head turned away as an 'inattentive' subject. Goats, as animals with laterally positioned eyes, might still be able to pay attention to events and subjects even though their head is averted from those. Thus, head orientation might not be a reliable cue of attention for goats and other ungulates. This explanation is partially in line with the finding that goats can use human pointing gestures, but not human head orientation, to find a hidden food reward (*Kaminski et al., 2005*; *Nawroth, von Borell & Langbein, 2015*).

The first two experiments showed that goats evaluate the body orientation of the experimenter when approaching a human or choosing between two experimenters. A control test showed that goats do not simply prefer to approach the experimenter whose hands were visible. Our results validate previous findings in similar experiments on goats (*Nawroth, von Borell & Langbein, 2015*; *Nawroth, Brett & McElligott, 2016*), which show

that goats alter their behaviour depending on human body orientation. Body orientation is a more salient cue than head orientation alone (*Emery, 2000*). Similar results have been obtained for dogs (*Call et al., 2003*) and primates (*Flombaum & Santos, 2005*; *Bourjade et al., 2014*).

Although goats did not alter their behaviour depending on head orientation, they preferred to approach a human that was not covering her face behind a blind. This indicates that, although the head direction might not be taken into account when making a choice, the human head itself conveys crucial information for goats (*Tate et al., 2006*). However, goats did not prefer to approach the person that was not covering her eyes with small blinds. It is possible that this lack of preference might be because both experimenters partially covered their faces with the small blinds (eyes vs mouth), and thus leading to an indifferent choice in our test subjects.

Goats preferred to approach humans that had their eyes open vs closed, while the visibility vs non-visibility of the area of the eyes did not alter their behaviour. This is a rather unexpected result, given their insensitivity to the head orientation in the same experiment. Several primate species (*Flombaum & Santos, 2005*; *Bourjade et al., 2014*), but also dogs (*Udell, Dorey & Wynne, 2011*) and horses (*Proops & McComb, 2010*), have been found to be sensitive to open vs closed eyes of human. However, all these studies showed also that subjects considered the human head orientation as crucial when attributing attention. Goats might also have been sensitive to the eyes closed, but not to the eyes covered, because the previous reflects a more naturalistic setting. Future studies need to further examine when and how evaluate eye visibility.

In conclusion, our results provide evidence that species domesticated for production are sensitive to subtle human cues, such as whether eyes are open or closed. Our lack of evidence for goats' interpretation of human head orientation as a cue to their attentional stance demands further evaluation. We propose that ontogenetic factors such as previous experience with humans might play a crucial role in the development of human-directed behaviour.

## ACKNOWLEDGEMENTS

We thank Luigi Baciadonna, Jemma Mary Brett, Pamela Prentice and Fang-Ting Wu for help during data collection, and the editor and the two reviewers for their very helpful comments on the manuscript. We also thank Robert Hitch and all the employees and volunteers of Buttercups Sanctuary for Goats (http://www.buttercups.org.uk) for their excellent help and free access to the animals.

### Funding

This work was supported by grants from the Deutsche Forschungsgemeinschaft (NA 1233/1-1) to C. Nawroth, and a grant from the Farm Sanctuary 'The Someone Project' to A.G. McElligott and C. Nawroth. The funders had no role in study design, data collection and analysis, decision to publish or preparation of the manuscript.

## Grant Disclosures

The following grant information was disclosed by the authors:
Deutsche Forschungsgemeinschaft: NA 1233/1-1.
Farm Sanctuary 'The Someone Project'.

## Competing Interests

The authors declare that they have no competing interests.

## Author Contributions

- Christian Nawroth conceived and designed the experiments, performed the experiments, analysed the data, wrote the paper and prepared figures and/or tables.
- Alan G. McElligott conceived and designed the experiments, wrote the paper.

## Animal Ethics

The following information was supplied relating to ethical approvals (i.e. approving body and any reference numbers):

The study was approved by the Animal Welfare and Ethical Review Board committee of Queen Mary University of London (Ref. QMULAWERB032015).

## Data Deposition

The raw data has been supplied as Supplemental Dataset Files.

## Supplemental Information

Supplemental information for this article can be found online at http://dx.doi.org/10.7717/peerj.3073#supplemental-information.

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
