# Peer review of "Human head orientation and eye visibility as indicators of attention for goats (Capra hircus)"

_PeerJ, doi:10.7717/peerj.3073_

## Round 0.1 · original submission · Major Revisions

It is especially crucial that you address the reviewers' concerns regarding the methods.

Line 134 features this sentence: “During both conditions, the experimenter’s hands were oriented away from the focal subject to avoid cueing.” Please describe this hand orientation in detail.

·

Basic reporting

This article is well written using clear and unambiguous text. It meets professional standards both of quality of writing and by providing sufficient introduction to the work. The background provided in the literature review demonstrates how the work fits into the broader scientific literature quite nicely. Further, the article structure is as expected in a psychology report and the authors have included their data in a format that can be easily understood by others. Finally, the work is 'self contained' -- it represents an appropriate publication and includes all relevant results.

However, I think the authors need to include more information in their figure captions. For example, Figure 1 is confusing. Black lines indicate ‘threshold’ lines — does that mean in A that the goat had to cross the midline of the researcher in order to be considered approaching him/her? That makes sense in the case of the ‘back’ condition but not in the case of the ‘front’ condition. Ditto in figure B — must the goat have gone in front of the ‘head away’ person to count? What if they went to the same position just south of the person so that they were standing right in front of the person’s face? All of this is cleared up later in the text but the figure caption as it is right now is not enough to make this clear.

Experimental design

This work fits within the aims and scope of PeerJ and it is clear what the author's research question is and how it fills a knowledge gap. Namely, the authors seek to explore whether animals domesticated for production (goats) show similar social cognition skills as animals domesticated for companionship / to work closely with humans (i.e., dogs). For the most part it seems this work is rigorous. However, there are some places where there is not sufficient detail to assess or to be reproducible by another investigator. I detail these concerns below.

The scoring procedure is ambiguous — “approach behavior of the goats was scored live. After a goat approached the experimenter to approximately 1 m, the experimenter scored the behavior… “ Was the experimenter being approached the one who scored the behavior of the goat? For the ‘back approach’ the goat would necessarily have had to approach the front first, correct? Before it crossed the line? How was the scored? Was only the final position after 5 seconds scored? This part is very confusing.

I'm also confused about the data and the analysis itself -- why did the authors use Fischer’s test in E1 and a binomial test in E2? Shouldn’t both have been binomial? The data from E1.2 could not have been analyzed with a binomial test, but I didn't know that until I looked at the actual data set. I think the authors need to be much more clear in the text about what their data actually looked like for the different experiments. Having the attached data file is certainly helpful, but readers should not have to consult that in order to be able to understand what the data is.

The authors mention interrater reliability in Experiment 3, but what about in experiments 1 and 2? Who got to decide what the goats did? In experiment 3, what about in the cases when it was ambiguous what the goats did? Did they all agree it was ambiguous?

Methodologically, Experiment 3 needs a little more detail. Who called “Come here?” in the case of the two experimenters? Was it both of them at the same time? I also found the crossing hands over confusing -- perhaps including this in the figure would help eliminate that confusion. Also a little more information about the motivational trials would be helpful here -- for example, who gave the goats the motivational trial between test trials?

Validity of the findings

The data appears robust but without some of the clarification of the methods described above I can't comment on whether they are controlled appropriately. Conclusions are well stated and limited to supporting results. I would, in fact, welcome some additional speculation in the discussion section. It seems the authors explain each individual finding ok, but not the overall pattern — why, for example, do the authors think that the goats sometime understand the importance of faces but not other times? Ditto with the gots understanding of the importance of eyes -- why do they show some conflicting evidence here? I would welcome the authors to provide a little more speculation about that here.

Additional comments

One quick minor comment -- there's some awkward phrasing on line 399 — ‘although the human head has not to be in the direction of the goat’ This is a bit ambiguous and leads to a couple different interpretations depending on how it's read. Perhaps being a little clearer here will help.

Overall, I really liked this study. I think the authors just need to include a little more detail.

Reviewer 2 ·

Basic reporting

The paper is well written and the English is of a good standard. There are a couple of minor errors detailed below:
L30. should read "for approaching two..."
L58. should read "However, researchers did not find..."
L77. since the experiment mentioned here was conducted by the present authors, please change this sentence to "In these experiments..."
L93. please change to "had the opportunity to approach either an attentive..."
L399. please change to "and are used to receiving food directly, although the human head is not necessarily in the direction of the goat at the time"

Literature references are generally appropriate, I'm not sure their ability to recognise offspring calls and to perform well on a puzzle box tasks is directly relevant to this research (Briefer et al. 2012 & 2014). The authors could also have mentioned the use of head and eye gaze in relevant object choice tasks – for example goats have been shown not to use head orientation. Other production animals have also been tested such as pigs, so this is not such an understudied group of animals. Furthermore, a large number of wild species have also been studied (some are mentioned in the ms), so the abilities of species not selected for human companionship have been explored.

Structure, figures etc are all of a professional standard, raw data are provided. The results are appropriate to the hypotheses.

In experiment 3, to further assess if there were any behavioural differences, perhaps the authors could assess response latency, perhaps the goats were more hesitant when approaching people with their head averted (and the other conditions). Also, did the goats that approached the head averted experimenter across all experiments show a higher level of tactile or auditory behaviours, perhaps indicative of sensitivity to inattention?

Experimental design

The research falls within the scope of the journal. The research questions are clearly defined and how this study fits in to existing literature is explained (but see comment above). There are no ethical issues with the study.

It would have been preferable to record all the experiments for future reference and second coding. Were the results recorded by a second experimenter or the experimenter performing the study?

A few more details of the methods would be beneficial. Firstly, were the same goats used across the experiments? And were the goats rewarded in experiments 1 and 2? For “correct” and “incorrect” responses? If all responses were rewarded, was the delay to reward the same for inattentive and attentive choices? Goats that were rewarded for approaching an attentive person in experiment 1 may have learnt to discriminate in experiment 3. If goats were involved in more than study, perhaps the potential effect of this could be explored statistically. In order to confirm the goats had little or no prior training, please outline the previous experiments that the goats had participated in.

Some of the methods seem a little imprecise. How did the experimenters measure the approximate distances of the goats? In experiment 2.1 how was “about 90 degrees” determined. Again, it is a shame that these trials were not videoed so that more accurate coding could have been done at a later date, with additional second coding. Who determined whether there was a “side approach”?

Although there was not always a difference seen in the trials, across all studies the goats were choosing between a still and moving person. The method would have been more standardized if either both people moved in to position as the goat approached or both were static at the start of the trials.

How did the experimenters ensure that they called the goat at the same time?

Validity of the findings

The statistics used are appropriate – see comment above for suggestions of additional analyses.

In experiments 1 & 2, in which conditions did the goats fail to respond?

L288. Additional controlled experimentation is welcomed but is there any reason to believe environmental disturbances or conspecifics would have biased the responses in any particular direction?

L408. Yes, but one might expect that head cues are more salient than eye cues.

In the head averted condition, one eye is still visible (side view). For animals with laterally placed eyes, such as goats, they may well be able to still be attentive to objects even when their head is reasonably averted from the subject, thus head orientation may not be a reliable cue to attention in conspecifics and perhaps this knowledge is transferred to interactions with humans.

---

## Round 0.2 · accepted · Accept

You have done an excellent job addressing the reviewers' concerns.

·

Basic reporting

no comment

Experimental design

no comment

Validity of the findings

no comment

Additional comments

The authors addressed my concerns in this manuscript. I think it should be published with these revisions.

One typo -- line 85" "nor head direction" should read "not head direction"

Reviewer 2 ·

Basic reporting

I am happy with the revised paper.

Experimental design

See above comment.

Validity of the findings

See above comment.

Additional comments

See above comment.